# Efficient Algorithms for Coded Multicasting in Heterogeneous Caching Networks

**DOI:** 10.3390/e21030324

**Published:** 2019-03-25

**Authors:** Giuseppe Vettigli, Mingyue Ji, Karthikeyan Shanmugam, Jaime Llorca, Antonia M. Tulino, Giuseppe Caire

**Affiliations:** 1Department of Electrical Engineering and Information Technology (DIETI), Universitá di Napoli Federico II, 80138 Napoli, Italy; 2Department of Electrical and Computer Engineering (ECE), University of Utah, Salt Lake City, UT 84112, USA; 3IBM Research, New York, NY 10598, USA; 4Department of Math and Algorithms, Nokia Bell Labs, Murray Hill, NJ 07738, USA; 5Faculty of Electrical Engineering and Computer Science (EECS), Technical University of Berlin, 10587 Berlin, Germany

**Keywords:** caching networks, random fractional caching, coded caching, coded multicasting, index coding, finite-length analysis, graph coloring, approximation algorithms

## Abstract

Coded multicasting has been shown to be a promising approach to significantly improve the performance of content delivery networks with multiple caches downstream of a common multicast link. However, the schemes that have been shown to achieve order-optimal performance require content items to be partitioned into several packets that grows exponentially with the number of caches, leading to codes of exponential complexity that jeopardize their promising performance benefits. In this paper, we address this crucial performance-complexity tradeoff in a heterogeneous caching network setting, where edge caches with possibly different storage capacity collect multiple content requests that may follow distinct demand distributions. We extend the asymptotic (in the number of packets per file) analysis of shared link caching networks to heterogeneous network settings, and present novel coded multicast schemes, based on *local graph coloring*, that exhibit polynomial-time complexity in all the system parameters, while preserving the asymptotically proven multiplicative caching gain even for finite file packetization. We further demonstrate that the packetization order (the number of packets each file is split into) can be traded-off with the number of requests collected by each cache, while preserving the same multiplicative caching gain. Simulation results confirm the superiority of the proposed schemes and illustrate the interesting request aggregation vs. packetization order tradeoff within several practical settings. Our results provide a compelling step towards the practical achievability of the promising multiplicative caching gain in next generation access networks.

## 1. Introduction

Recent information-theoretic studies [1,2,3,4,5,6,7,8,9,10,11,12,13,14,15,16,17,18,19,20,21,22,23,24,25,26,27,28,29,30,31,32,33,34,35,36,37,38,39,40,41,42,43,44,45,46,47,48,49] have characterized the fundamental limiting performance of several caching networks of practical relevance, in which throughput scales linearly with cache size, showing great promise to accommodate the exponential traffic growth experienced in today’s communication networks [50]. In this context, a caching scheme is defined in terms of two phases: the *cache placement phase*, which operates at a large time-scale and determines the content to be placed at the network caches, and the *delivery phase*, during which user requests are served from the content caches and sources in the network. Some of the network topologies studied include shared link caching networks [1,2,8,9,10,11,12,13,14], device-to-device (D2D) caching networks [17,18,19,33,34], hierarchical caching networks [24], multi-server caching networks [29], and combination caching networks [36,37,38,39,40,41,42].

Consider a network with one source (e.g., base station) having access to *m* files, and *n* users (e.g., small-cell base stations or end user devices), each with a cache memory of *M* files. In [17], the authors showed that if the users can communicate between each other via D2D communications, a simple distributed random caching policy and TDMA-based unicast D2D delivery achieves the order-optimal throughput $\Theta \left(\max\{\frac{M}{m},\frac{1}{m},\frac{1}{n}\}\right)$ whose linear scaling with *M* when Mn≥m exhibits a remarkable multiplicative caching gain, in the sense that the per-user throughput grows proportionally to the cache size *M* for fixed library size *m*, and it is independent of the number of users *n* in the system. Moreover, in this scheme each user caches entire files without the need for partitioning files into packets, and missing files are delivered via unicast transmissions between neighbor nodes, making it efficiently implementable in practice. We recall that order-optimality refers to the fact that the multiplicative gap between information-theoretic converse and achievable performance can be bounded by a constant number when m,n→∞.

In the case that users cannot communicate between each other, but share a multicast link from the content source, the authors in [8,9] showed that the use of coded multicasting (also referred to as index coding [51]) allows achieving the same order-optimal worst-case throughput as in the D2D caching network. In this case, however, in order to create enough coding opportunities during the delivery phase, requested files are required to be partitioned into a number of packets that grows exponentially with the number of users, leading to coding schemes of exponential complexity [8,9,21].

In [10,12], the authors considered the same shared link caching network, but under random demands characterized by a probability distribution, and proposed a scheme consisting of random aggregate popularity (RAP) placement and chromatic number index coding (CIC) delivery, referred to as RAP-CIC, proved to be order-optimal in terms of average throughput. The authors further provided optimal average rate scaling laws under Zipf [52] demand distributions, whose analytical characterization required resorting to a polynomial-time approximation of CIC, referred to as greedy constrained coloring (GCC). Using RAP-GCC, the authors further established the regions of the system parameters, in which multiplicative caching gains are potentially achievable. While GCC exhibits polynomial complexity in the number of users and packets, the order-optimal performance guarantee still requires, in general, the packetization order (number of packets per file) to grow exponentially with the number of users, as showed in [21].

It is then key to understand if the promised multiplicative caching gain, shown to be asymptotically achievable by the above-referenced schemes, can be preserved in practical settings of finite packetization order. In this context, we shall differentiate between coded multicast schemes that assume a deterministic vs. a random cache placement phase. Deterministic placement policies determine where to store file packets according to a deterministic procedure that takes into account the ID of each packet. In contrast, random placement policies, after determining the number of packets to be cached of each file at each cache, choose the exact packet IDs uniformly at random. While the increased structure of deterministic placement policies can be exploited to design more efficient coded multicast algorithms, random placement policies are desirable in practice, as they provide increased robustness by requiring less cache configuration changes under system dynamics.

The seminal work of [21] showed that all previously proposed schemes (based on both deterministic and random cache placement) required exponential packetization, and that under random placement, no graph-coloring-based coded multicast algorithm can achieve multiplicative caching gains with sub-exponential packetization. Since the fundamental results of [21], several works have studied the now central problem in caching of finite file packetization. The authors in [53] connect the caching problem to resolvable combinatorial designs and derive a scheme that while improving exponentially over previous schemes [8,9,21], still requires exponential packetization. In [54], the authors introduce the combinatorial concept of Placement Delivery Array (PDA) and derive a caching scheme where the packetization scales super-polynomially with the number of users. The work in [22] establishes a connection with the construction of hypergraphs with extremal properties, and provides the first sub-exponential (but still intractable) scheme. Somewhat surprisingly, some of the authors of [21] introduced a new combinatorial design based on Ruzsa-Szeméredi graphs in [30] and showed that a linear scaling of the number of packets per file with *n* can be achieved for a throughput of $\Theta (n^{-\delta })$, where δ can be arbitrarily small. However, all the above studies focus on coded multicast algorithms that assume a deterministic cache placement phase. Under random cache placement, several coded multicast algorithms have been proposed in the context of homogenous shared link caching networks [55,56,57,58,59,60], including our previous work that serves as the basis for this paper.

In this work, we address the important problem of finite-length coded multicasting under random cache placement, focusing on a more general heterogeneous shared link caching network, in which caches with possibly different sizes collect possibly multiple requests according to possibly different demand distributions (see Figure 1). As shown in Figure 1, this scenario can be motivated by the presence of both end user caches and cache-enabled small-cell base stations or WLAN access points sharing a common multicast link. In this case, each small-cell base station can be modeled as a user cache placing multiple requests. In addition, multiple requests per user also arise in the presence of delay-tolerant content requests (e.g., file downloading). While there have been several information-theoretic studies of shared link caching networks with distinct cache sizes [61,62,63], and with multiple per-user requests [13,14,34,64,65], none of these works considered the finite-length regime nor addressed the joint effect of random demands, heterogenous cache sizes, and multiple per-user requests.

The contributions of this paper are as follows:We provide a generalized model for heterogeneous shared link caching networks, in which users can have different cache sizes and make different number of requests according to different demand distributions.We design two novel coded multicast algorithms based on *local graph coloring*, referred to as Greedy Local Coloring (GLC) and Hierarchical Greedy Local Coloring (HgLC) that exhibit polynomial-time complexity in both the number of caches and the packetization order. In combination with the Random Aggregate Popularity (RAP) placement policy of [10,12], we show that the overall schemes RAP-GLC and RAP-HgLC are order-optimal in the asymptotic file-length regime.Focusing on the finite-length regime, in which content items can be partitioned into a finite number of packets, we show how the general advantage of local graph coloring is especially relevant when the number of per-user requests grow. We validate via simulations the superiority of RAP-GLC, especially with high number of per-user requests. We then show how RAP-HgLC, with a slight increase in the polynomial complexity order, further improves the caching gain of RAP-GLC, remarkably approaching the multiplicative gain that existing schemes can only guarantee in the asymptotic file-length regime.We demonstrate that there is a tradeoff between the required packetization order and the number of requested files per user. In particular, for a given target gain, if the number of requests increases, then the number of packets per file can be reduced, while preserving the target gain. We further quantify the regime of per-user requests for which a caching scheme with unit packetization order (i.e., a scheme that treats only whole files) is order-optimal. Our analysis illustrates the key impact of content request aggregation in time and space on caching performance. That is, if edge caches can wait for collecting multiple requests over time and/or aggregate requests from multiple users, the same performance can be achieved with lower packetization order, and hence lower computational complexity.

The paper is organized as follows. Section 2 introduces the network model and problem formulation. Section 3 describes the construction of coded multicast algorithms using graph coloring, with special focus on the advantages of local graph coloring. Section 4 presents novel polynomial-time local-graph-coloring-based coded multicast schemes. Section 5 analyzes the effect of request aggregation on the performance–complexity tradeoff. Section 6 presents simulation results and related discussions. Finally, concluding remarks are given in Section 7.

## 2. Network Model and Problem Formulation

We consider a caching network formed by a source node with access to a content library, connected to several caching nodes/users via a single shared (multicast) link. Similar to previous works [8,9,10,12,13,14,21,22,30], we define a caching scheme in terms of two phases:*Placement phase*, which operates at a large time-scale and determines the content to be placed at the caching nodes,*Delivery phase*, during which users requests are served from the content caches and sources in the network.

However, differently from previous works, we generalize the model to a heterogeneous system in which each caching node has a possibly different cache size and requests a possibly different number of files. A practical example of our setting can be represented by a macro base station connected to several cache-enabled small-cell base stations, and a number of user devices served either by the macro base station or by the small-cell base stations. In this setting, each small cell acts as a super user requesting multiple files resulting from the requests of the users it serves.

Specifically, the heterogeneous caching network consists of a single source node storing a library of files F={1,⋯,m}, each with entropy *F* bits, and *n* user nodes U={1,⋯,n}, each with a cache of storage capacity $M_{u}F$ bits (i.e., each user caches up to $M_{u}$ files). Each user *u* can requests $L_{u}$$\left(1\leq L_{u}\leq m\right)$ different files according to its individual request probability distribution. We assume that the library files have finite length and consequently a finite packetization order. Our main objective is to design a caching scheme that minimizes the number of transmissions required to satisfy the demands of all users.

In a homogeneous network setting with infinite packetization order, recent works [8,9,10,12,13,14] have shown that it is possible to satisfy a scaling number of users with only a constant number of multicast transmissions. The achievable schemes configure user caches with complementary (side) information during the caching phase, such that the resulting coded multicasting opportunities that arise during the delivery phase can be used to minimize the transmission rate (or load) over the shared multicast link. Specifically, reference [12] showed that under Zipf file popularity, a properly optimized random fractional placement policy, referred to as Random Aggregate Popularity (RAP) caching, achieves order-optimality when combined with a graph-coloring-based coded multicast scheme. Unfortunately, even in the homogenous setting, it was shown in [21] that a central limitation of all previous works is that they require infinite packetization order: all existing caching schemes achieve at most a factor of two gain when the packetization order is finite.

In this work, inspired by the fundamental throughput-delay-memory tradeoff derived in [21], our goal is to design computationally efficient schemes that provide good performance in the finite packetization regime. For the caching phase, (1) we restrict our placement policies to the class of random fractional schemes described in [9,10,12,13,14], proved to be order-optimal in the homogeneous setting. For the delivery phase, (2) we focus on the class of graph-coloring-based index coding schemes, and design two novel polynomial-time algorithms that employ local graph coloring on the (index coding) conflict graph [51].

### 2.1. Random Fractional Cache Placement

The class of random fractional placement schemes is described as follows:Packetization: Each file is partitioned into *B* packets of equal-size F/B bits, where the integer *B* is referred to as the packetization order. Each packet is represented by a symbol in finite field $F_{2^{F/B}}$, where we assume that F/B is large enough.Random Placement: Each user *u* caches $p_{f,u}M_{u}B$ packets independently at random from each file *f*, where $p_{f,u}$ is the probability that file *f* is cached at user *u*, and satisfies $0\leq p_{f,u}\leq 1/M_{u},\forall f\in F$ such that $\sum_{f=1}^{m}p_{f,u}=1,\forall u\in U$.

We introduce a *caching distribution* matrix $P=\left[p_{f,u}\right]\in R_{+}^{m\times n}$, where f∈F and u∈U. Please note that the number of packets of file *f* cached at user *u*, $p_{f,u}M_{u}B$, can be directly determined from the caching distribution matrix P. As described in [10,12,13,14], the caching distribution must be properly optimized to balance the gains from local cache hits (where requested packets are served by the local cache) and coded multicast opportunities (where requested packets are served by coded transmissions that simultaneously satisfy distinct user requests). When this is the case, we refer to the cache placement scheme as Random Aggregate Popularity (RAP) caching (see e.g., [10,12,13,14]). Given the number of packets to be cached of a given file, the actual indices of the packets to be cached are chosen uniformly at random, and independently across users. We use $C_{u,f}$ to denote the set of packets of file *f* cached at user *u* and $C=\{C_{u,f}\}$ with u∈U and f∈F to denote the *packet-level* cache placement realization.

The goal of the placement phase is to configure the user caches to create coding opportunities during the delivery phase that allow serving distinct user requests via common multicast transmissions. Compared to deterministic placement [8], random placement schemes allow configuring user caches with lower complexity and increased robustness, i.e., changes in system parameters (e.g., number of users, number files, file popularity) require less changes in users’ cache configurations [12].

Recall that the placement phase operates at a much larger time-scale than the delivery phase, and hence is unaware of the requests in the subsequent delivery rounds. Therefore, the placement phase can be designed according to the demand distribution, but must be independent of the requests realizations.

### 2.2. Random Multiple Requests

Each user u∈U requests $L_{u}$$\left(1\leq L_{u}\leq m\right)$ files independently from other users, following a probability distribution $q_{f,u}$ with $q_{f,u}\in \left[0,1\right]$ and $\sum_{f=1}^{m}q_{f,u}=1$ (i.e., for each request of user *u*, file *f* is chosen with probability $q_{f,u}$). We introduce a *demand distribution* matrix $Q=\left[q_{f,u}\right]\in R_{+}^{m\times n}$, where f∈F and u∈U. In the following, we use $W=\{W_{u,f}\}$, with u∈U and f∈F, to denote the *packet-level* demand realization where $W_{u,f}$ denotes the packets of file *f* requested by user *u*.

The multiple-request parameters $\left\{L_{u}\right\}$ have a key operational meaning, in that it captures the possibility of edge caches to collect requests across time and space. That is, $L_{u}$ may represent the amount of requests collected over time (given the delay tolerance of some content requests) as well as the amount of requests collected across space from users served by the given edge cache (e.g., when edge caches are located at helper nodes or small-cell base stations serving multiple individual users).

### 2.3. Performance Metric

For given realizations of the random fractional cache placement and the random multiple requests, the goal is to design a delivery scheme that minimizes the rate over the shared multicast link required to satisfy all user requests. Since one placement phase is followed by an arbitrarily large number of delivery rounds (each characterized by a new independent request realization), the rate (or load) of the system refers only to the delivery phase (i.e., asymptotically the cache placement costs no rate). Furthermore, it makes sense to consider the average rate, where averaging with respect to the users request distribution takes on the meaning of a time-averaged rate, invoking an ergodicity argument.

At each request round, let $F=\{f_{1},f_{2},\cdots ,f_{n}\}$ be the demand realization, where $f_{u}=\left\{f_{1,u},f_{2,u},\cdots ,f_{L_{u},u}\right\},u\in U$. The source node computes a multicast codeword as a function of the library and the demand realization F. We assume that the source node communicates to the user nodes through an error-free deterministic shared multicast link.

Given the demand realization F, let the total number of bits transmitted by the source node be J(F). We are interested in the average performance of the coded multicast scheme, and hence define the average rate (or load) as the number of transmitted bits normalized by the file size:(1)$$R=\frac{E\left[J(F)\right]}{F},$$
where the expectation is over the random demand distribution.

## 3. Graph-Coloring-Based Coded Multicast Delivery

It is important to note that for given cache placement and demand realizations, the delivery phase of a caching scheme reduces to an index coding problem *with a twist*. The only difference with the conventional index coding problem introduced in [51] is that the cache information may contain *part of* (as opposed to entire) requested files, and that users may request *multiple* (as opposed to single) files. Nevertheless, as in index coding, the problem can still be represented by a *conflict graph* [10,12,13,14], where vertices represent requested packets, and an edge between two vertices indicates a conflict, in the sense that the packet represented by one vertex is not present in the cache of the user requesting the packet represented by the other vertex. By construction, packets with no conflict in the graph can be simultaneously transmitted via an XOR operation. Performing graph coloring on the conflict graph and transmitting the packets via proper XOR operations, according to the graph coloring, results in an achievable linear index coding scheme, which we refer to as a coded multicast scheme.

In the following, we first illustrate how to construct the conflict graph, we then review classical linear index coding schemes, and then describe our proposed graph-coloring-based coded multicast schemes.

### 3.1. Conflict Graph Construction

Given cache placement realization C and demand realization W, the directed conflict graph $H_{C,W}^{d}=\left(V,E\right)$ can be constructed as follows:Vertices: For each packet request in W, there is a vertex in $H_{C,W}^{d}$. Each vertex v∈V is uniquely identified or labeled by a *packet-user* pair {ρ(v),μ(v)}, where ρ(v) denotes the identity of the packet, and μ(v) the user requesting it. Hence, if a packet is requested by multiple users, such a packet is represented in as many vertices as the number of users requesting it. Such vertices have the same packet label ρ(v), but different user label μ(v).Arcs: For any $v_{1},v_{2}\in V$, there is an edge $(v_{2},v_{1})\in E$ with direction from $v_{2}$ to $v_{1}$ if and only if $\rho \left(v_{1}\right)\neq \rho \left(v_{2}\right)$ and packet $\rho (v_{1})$ is not in the cache of user $\mu (v_{2})$.

To better understand the rationale behind the conflict graph and its construction, note that for any two vertices $v_{1}$ and $v_{2}$ that are labeled as $\left\{\rho \left(v_{1}\right),\mu \left(v_{1}\right)\right\}$ and $\left\{\rho \left(v_{2}\right),\mu \left(v_{2}\right)\right\}$, respectively, we have the following three possible cases:$\rho \left(v_{1}\right)\neq \rho \left(v_{2}\right)$ and $\mu \left(v_{1}\right)=\mu \left(v_{2}\right)$: This indicates that two different packets are requested by the same user. Then, $v_{1}$ and $v_{2}$ are mutually conflicting, in the sense that if sent within the same time-frequency resource they interfere with each other. Hence, in the conflict graph, they are connected with two directed edges, $(v_{1},v_{2})\in E$ and $(v_{2},v_{1})\in E$;$\rho \left(v_{1}\right)=\rho \left(v_{2}\right)$ and $\mu \left(v_{1}\right)\neq \mu \left(v_{2}\right)$: This indicates that the same packet is requested by two different users. Then, $v_{1}$ and $v_{2}$ are not conflicting, and hence not connected in the conflict graph; i.e., $(v_{1},v_{2})\notin E$ and $(v_{2},v_{1})\notin E$;$\rho \left(v_{1}\right)\neq \rho \left(v_{2}\right)$ and $\mu \left(v_{1}\right)\neq \mu \left(v_{2}\right)$: This indicates that two different packets are requested by two different users. In this case, if packet $\rho (v_{1})$ is in the cache of user $\mu (v_{2})$, then, even if $\rho (v_{1})$ and $\rho (v_{2})$ are sent within the same time-frequency resource, user $\mu (v_{2})$ will not suffer from interference, since, using its cache information, it can cancel out the undesired packet $\rho (v_{1})$ from the received signal. On the other hand, if packet $\rho (v_{1})$ is not in the cache of user $\mu (v_{2})$, then $v_{1}$ conflicts with $v_{2}$, and a directed edge is drawn from $v_{2}$ to $v_{1}$. Similarly, $(v_{1},v_{2})\in E$ if and only if $\rho \left(v_{2}\right)\notin C_{\mu (v_{1})}$.

Based on the above construction, it follows that the number of interference dimensions faced by a given node is at most the number of its outgoing neighbors.

To illustrate the construction of the directed conflict graph $H_{C,W}^{d}$, we present the following example.

**Example** **1.**
*We consider a network with n=3 users denoted as U={1,2,3} and m=3 files denoted as F={A,B,C}. We assume $M_{u}=1,\forall u\in U$ and partition each file into three packets. For example, $A=\{A_{1},A_{2},A_{3}\}$. Let $p_{A,u}=p_{B,u}=p_{C,u}=\frac{1}{3}$ for u∈U, which means that one packet from each of A,B,C is stored in each user’s cache. For the sake of notational convenience, we assume a symmetric caching realization, where the caching configuration C is given by $C_{u,A}=\left\{A_{u}\right\},C_{u,B}=\left\{B_{u}\right\},C_{u,C}=\left\{C_{u}\right\}$). That is, the cache configuration of each user u∈U is $C_{u}=\left\{A_{u},B_{u},C_{u}\right\}$. We let each user make two requests, i.e., $L_{u}=2$(∀u∈U). Specifically, we let user *1* request A,B, user *2* request B,C, and user *3* request C,A, i.e., $f_{1}=\left\{A,B\right\},f_{2}=\left\{B,C\right\},f_{3}=\left\{C,A\right\}$), such that $W_{1}=\left\{A_{2},A_{3},B_{2},B_{3}\right\},W_{2}=\left\{B_{1},B_{3},C_{1},C_{3}\right\},W_{3}=\left\{A_{1},A_{2},C_{1},C_{2}\right\}$. The associated directed conflict graph is shown in Figure 2.*


### 3.2. Code Construction

Let $\omega_{v}$ denote the content (or realization) of packet ρ(v), v∈V, represented by a symbol in $F_{q}$. In general, in a linear index coding scheme of length ℓ, every vertex *v* is associated with a “coding" vector $g_{v}\in F_{q}^{\ell \times 1}$ where v∈[1:|V|]. Let $G=[g_{1},\cdots g_{\left|V\right|}]$ and $\omega ={\left[\omega_{1},\cdots ,\omega_{\left|V\right|}\right]}^{T}$. Then, the transmitted codeword, $x\in F_{q}^{\ell \times 1}$, is built as follows:(2)$$\begin{array}{c} x=\sum_{v\in V}\omega_{v}g_{v}=G\omega , \end{array}$$

Let N(v)={w:(v,w)∈E} be the out-neighborhood of *v*. For any feasible scalar linear index coding scheme of the form (2), the following *interference alignment* condition is necessary: For every vertex *v*, the coding vector $g_{v}$ should be linearly independent of all the coding vectors assigned to the out-neighborhood of *v*.

In the following, we describe how to construct coding vectors satisfying the interference alignment condition for every vertex. For ease of notation, we use $H^{d}$ to denote the directed conflict graph, and H to represent its underlying undirected skeleton, where the direction of edges is ignored. Recall that an undirected skeleton of a directed graph $H^{d}$ is an undirected graph where there is an undirected edge between $v_{1}$ and $v_{2}$ if, between $v_{1}$ and $v_{2}$, there is a directed edge in either or both directions in $H^{d}$.

#### 3.2.1. Graph Coloring and Chromatic Number

A well-known procedure to construct the coding vectors $\left\{g_{v},\;v\in N\left(v\right)\right\}$ is the *coloring* of $H^{d}$. In the following, when used without any qualification, a coloring of a directed graph is considered to be a proper (vertex) coloring of its underlying undirected skeleton H, where a proper coloring is a labeling of the graph’s vertices with colors, such that no two vertices sharing the same edge have the same color. Please note that by definition, any subset of nodes with the same color in a proper coloring form an *independent set* (i.e., a subset of nodes in a graph, no two of which share the same edge). A coloring using at most *k* colors is called a (proper) *k*-coloring. The smallest number of colors needed in a proper coloring of $H^{d}$ is called its chromatic number, and is denoted by $\chi (H^{d})$. In the following, we explain why a coloring of $H^{d}$ provides a way to design the coding vectors $\left\{g_{v},\;v\in N\left(v\right)\right\}$. Let ξ be the total number of colors in a given coloring of $H^{d}$. Let $e_{i}$ be the *i*-th unit vector in the space $F_{q}^{\ell \times 1}$, with ℓ=ξ, i.e., $e_{i}={\left[0,0,\cdots ,1,\cdots ,0,0\right]}^{T}$, where the 1 is in the *i*-th position. Now, if vertex *v* is colored with color *i*, then, its coding vector is $g_{v}=e_{i}$. Making this choice for the coding vectors, the associated achievable rate is given by $\frac{\xi }{B}$. Since neighbors are assigned different colors, the interference alignment condition is satisfied for every vertex. Recalling the definition of $\chi (H^{d})$, it is immediate to see that the best achievable rate due to conflict graph coloring is given by $\frac{\chi (H^{d})}{B}$, and, according to the construction of the conflict graph, it is loosely bounded by:(3)$$\begin{array}{c} \sum_{f\in f_{u}}\left(1-p_{f,u}M_{u}\right)\leq \frac{\chi (H^{d})}{B}\leq \sum_{u=1}^{n}\sum_{f\in f_{u}}\left(1-p_{f,u}M_{u}\right), \end{array}$$
indicating that the achievable rate is a constant with regards to *B*. A much tighter bound will be given in Section 4.1.

#### 3.2.2. Local Graph Coloring and Local Chromatic Number

More efficient sets of coding vectors can be constructed using the approach proposed in [66], which exploits the direction information in $H_{C,W}^{d}$, resulting in the following advanced coding scheme:
**Definition 1** **(Local Coloring Number).***Given a proper coloring c of $H^{d}$, the associated local chromatic number is defined as:*(4)$$\begin{array}{c} \xi_{\mathrm{lc}}\left(c\right)=\max_{v\in V}\left|c\left(N^{+}\left(v\right)\right)\right| \end{array}$$*where $N^{+}\left(v\right)$ is the closed out-neighborhood of vertex v (i.e., vertex v and all its ongoing neighbors N(v)) and $\left|c(N^{+}\left(v\right))\right|$ is the total number of colors in $N^{+}\left(v\right)$ for a given proper color assignment c.*

The minimum local coloring number over all proper colorings is referred to as the *local chromatic number* and is formally defined as follows:
**Definition 2** **(Local Chromatic Number).***The directed local chromatic number of a directed graph $H^{d}$ is defined as:*(5)$$\begin{array}{c} \chi_{\mathrm{lc}}\left(H^{d}\right)=\min_{c\in C}\xi_{\mathrm{lc}}\left(c\right) \end{array}$$*where C denotes the set of all proper coloring assignments of $H^{d}$, $N^{+}\left(v\right)$ is the closed out-neighborhood of vertex v, and $\left|c(N^{+}\left(v\right))\right|$ is the total number of colors in $N^{+}\left(v\right)$ for a given proper color assignment c.*

**Encoding Scheme:** For a given realization of the cache placement (C) and user requests (W), let us consider the conflict graph $H_{C,W}^{d}$ as in Section 3.1. Given a (proper) ξ-coloring (i.e., a proper coloring of graph $H_{C,W}^{d}$ with ξ colors), we compute the associated local coloring number $\xi_{\mathrm{lc}}$. Set $\ell =\xi_{\mathrm{lc}}$ and p=ξ. Then, consider the columns of the generator H of an ℓ×p Maximum Distance Separable (MDS) [67] code over the field $F_{q}:q>p$. If the color of a vertex *v* is *i*, then the coding vector $g_{v}$ assigned to vertex *v* is given by *i*-th column $h_{i}$ of H. Then, the transmitted multicast codeword, $x\in F_{q}^{\ell \times 1}$, is given by (2).

**Decoding Scheme:** In any closed out-neighborhood, there are at most ℓ different colors (from the definition of local coloring). Since every ℓ columns of H are linearly independent (from the defining property of MDS codes), the coding vectors in any closed out-neighborhood have full rank, satisfying the interference alignment condition. The message $\omega_{v}$ at vertex *v* is obtained at user *v* as follows: (1) Using side information at user *v*, cancel out message parts corresponding to all vertices outside $N^{+}\left(v\right)$, i.e., $x^{\prime }=x-\sum_{u\notin N^{+}\left(v\right)}\omega_{u}g_{u}$. This is possible because, by the definition of the conflict graph $H^{d}$, the messages ${\left\{\omega_{u}\right\}}_{u\notin N^{+}\left(v\right)}$ are available as side information at user *v* and the encoding mechanism is known to all the users. (2) Find a vector z in the dual space of ${\left\{g_{u}\right\}}_{u\in N^{+}\left(v\right)\backslash \left\{v\right\}}$ such that $z^{T}x^{\prime }\neq 0$ (this is possible since $g_{v}$ is linearly independent of ${\left\{g_{u}\right\}}_{u\in N^{+}\left(v\right)\backslash \left\{v\right\}}$ because of the local chromatic number-based construction). Now, $z^{T}x^{\prime }=\left(z^{T}g_{v}\right)\omega_{v}$. Therefore, user *v* recovers its own message. It follows that all users can recover all the requested packets employing such linear scheme.

**Achievable Rate:** The coding scheme constructed as described above achieves a rate given by $\xi_{\mathrm{lc}}/B$, where *B* is the number of packets per file.

**Example** **2.**
*We consider an example shown in Figure 3. First, we assign colors to each vertex such that the total number of colors ξ=5, and count the local coloring number, which is $\xi_{\mathrm{lc}}=4$. Then, we construct the generator matrix A of a $\left(\xi =5,\xi_{\mathrm{lc}}=4\right)$ MDS code, which is given by*
(6)$$A=\left(\begin{array}{ccccc} 1 & 0 & 0 & 0 & 1\\ 0 & 1 & 0 & 0 & 1\\ 0 & 0 & 1 & 0 & 1\\ 0 & 0 & 0 & 1 & 1 \end{array}\right).$$

*After that, we assign the columns of A to $g_{v}$, corresponding from the left to the right to the vertices with the packets $\left\{A_{2},A_{3},B_{2},B_{1},A_{1}\right\}$, as shown in Figure 3. Finally, the transmitted codewords can be generated which are the rows of the right-hand side of *(2)**
(7)$$X\left(1\right)=A_{1}⊕A_{2},X\left(2\right)=A_{1}⊕A_{3}$$
(8)$$X\left(3\right)=A_{1}⊕B_{1},X\left(4\right)=A_{1}⊕B_{2}$$
*where the length of the code is $\xi_{\mathrm{lc}}/B=4/3$ file units. It can be easily verified that every user can decoded its desired packets with the cached ones.*


It is immediate to see that the best achievable rate due to local coloring is obtained by computing the local chromatic number of $H_{C,W}^{d}$ and using its associated coloring to design the coding vectors, yielding a rate $\chi_{\mathrm{lc}}/B$. However, note that to compute $\chi_{\mathrm{lc}}$, we must optimize over all proper colorings to find the local chromatic number. As with the chromatic number, this can be cast as an *Integer Program* and it is hence an NP-hard problem. To overcome this limitation, in Section 4, we propose a greedy approach that (i) exhibits polynomial-time complexity in all the system parameters, (ii) achieves close to optimal performance for finite packetization order, and (iii) is asymptotically (i.e., for infinite packetization order) order-optimal.

### 3.3. Benefits of Local Coloring

Consider the following relation established for general directed graphs in [66]:(9)$$\chi_{\mathrm{lc}}\left(H^{d}\right)\leq \chi \left(H^{d}\right)\leq \chi_{\mathrm{lc}}\left(H^{d}\right)O\left(\log n\right).$$

Focusing on the conflict graph of interest $H_{C,W}^{d}$, the number of vertices can be as large as $B\sum_{u\in U}L_{u}$. It then follows from (Equation 9) that the gap between the local chromatic number and the chromatic number can be as large as $\log(B\sum_{u\in U}L_{u})$. Please note that this multiplicative factor grows with the number of packets per file *B* and the number of per-user requests $L_{u}$, supporting the extra benefit of local coloring in the multiple-request scenario. In addition, the higher the number of per-user requests, the higher the directionality of the conflict graph, which is the main factor exploited by local coloring to reduce the achievable rate (see Section 3.2.2), further supporting the suitability of local coloring in increasingly practical settings where there is some form of spatial or temporal request aggregation.

## 4. Proposed Algorithms and Performance Analysis

As stated earlier, computing the local chromatic number is NP-hard. To circumvent this challenge, in this section, we propose two greedy coded multicast schemes, which together with the cache placement described in Section 2.1, yield the following two caching schemes: Randomized Aggregate Popularity-Greedy Local Coloring (RAP-GLC) and Randomized Aggregate Popularity-Hierarchical greedy Local Coloring (RAP-HgLC). In both cases, the steps for obtaining the coded multicast scheme are as follow:Given a realization of the cache placement (C) and of the user requests (W), build the conflict graph $H_{C,W}^{d}$ as in Section 3.1.Use any of the above algorithms (GLC or HgLC) to compute a proper coloring. Let ξ denote the number of colors used by either of the above algorithms to color $H_{C,W}^{d}$. Let $\xi_{\mathrm{lc}}$ be the associated local coloring number.Consider a $\left(\xi ,\xi_{\mathrm{lc}}\right)$ MDS code and compute the corresponding coded multicast scheme as described in Section 3.2.2.

### 4.1. Randomized Aggregate Popularity-Greedy Local Coloring (RAP-GLC)

The RAP-GLC algorithm generalizes the RAP-GCC (Random Aggregate Popularity-Greedy Constrained Coloring) algorithm introduced in [12]. RAP-GCC is a caching scheme based on random fractional caching for the placement phase and a coded multicast scheme built on greedy-graph-coloring -based linear index coding [51,68] for the delivery phase. RAP-GLC is more general than RAP-GCC in two aspects: (1) conventional coloring is replaced by local coloring to leverage possible gains in the multiple-request scenario, as described in Section 3.2.1 and Section 3.3, and (2) RAP-GLC adaptively (depending on the demand realization) chooses between naive or coded multicasting according to a threshold parameter, instead of sticking to one of them (as in RAP-GCC).

#### 4.1.1. RAP-GLC Algorithm Description

The algorithm associates to each vertex *v* a label or tag, composed of two fields i.e., $K_{v}\equiv \left(T_{D}\left(v\right),T_{C}\left(v\right)\right)$ with $T_{C}\left(v\right)$ denoting the subset of users caching the packet associated with vertex *v*, i.e.,
(10)$$T_{C}\left(v\right)\overset{\Delta }{=}\left\{u\in U:\rho \left(v\right)\in C_{u}\right\},$$
and $T_{D}\left(v\right)$ denoting the subset of users requesting the packet associated with vertex *v*, i.e.,
(11)$$T_{D}\left(v\right)\overset{\Delta }{=}\left\{u\in U:\rho \left(v\right)\in W_{u}\right\},$$
which includes the user itself μ(v) who requests ρ(v) and all the others requesting ρ(v). Please note that the cardinality of $T_{D}\left(v\right)$ indicates the popularity of packet ρ(v). Furthermore, let
$$T_{v}=\left\{\mu \left(v\right)\right\}\cup T_{C}\left(v\right).$$

Given a vertex *v*, if the cardinality of $T_{D}\left(v\right)$ is higher than a predetermined threshold parameter t∈{0⋯,n} i.e., $|T_{D}\left(v\right)|>t$, then all vertices $v^{\prime }$ such that $\rho \left(v\right)=\rho \left(v^{\prime }\right)$ are colored with the same color, leading to a naive multicast transmission scheme. If $|T_{D}\left(v\right)|\leq t$, then RAP-GLC greedily looks for a maximal set of vertices with the same $T_{v}$ (Algorithm 1, Line 14) and colors them with the same color if there is no conflict among the vertices (Algorithm 1, Line 15). The threshold parameter *t* is subject to optimization, as described in Section 4.1.2.

Doing this, RAP-GLC computes a valid coloring of the conflict graph H. Finally, the algorithm computes its associated local coloring number (Algorithm 1, Line 24). The coding scheme employed is based on the MDS code described in Section 3.2.1 associated with the above local coloring.

**Algorithm 1** RAP-GLC 1: Let C=∅; 2: Let c=∅; 3: **while**
V≠∅
**do** 4:  Pick an arbitrary vertex *v* in V; Let I={v}; 5:  Let V′=V\{v}; 6:  **if** { |TD(v)|>t } **then** 7:   **for all**
v′∈V′ with ρ(v′)=ρ(v)
**do** 8:    I=I∪v′; 9:   **end for**10:   Color all the vertices in I by c∉C;11:   Let c[I′]=c;12:   V=V\I′.13:  **else**14:   **for all**
v′∈V′ with Tv′≡Tv
**do**15:    **if** {There is no edge between v′ and I} **then**16:     I=I∪v′;17:    **end if**18:   **end for**19:   Color all the vertices in I by c∉C;20:   Let c[I]=c;21:   V=V\I.22:  **end if**23: **end while**24: **return** the local coloring number maxv∈V|c(N+(v))| and the corresponding color assignment c(N+(v)) for each *v*;

*Time Complexity:* In Algorithm 1, both the outer while-loop starting at Line 3, and the inner for-loop starting at Line 6 iterate at most |V| times, and all other operations inside the loops take constant time. Therefore, the complexity of RAP-GLC is $O(|V{|}^{2})$ or, equivalently, $O(n^{2}B^{2})$, since |V|≤nB, which is polynomial in |V| (or n,B).

#### 4.1.2. RAP-GLC Performance Analysis

In the following, we quantify the performance of RAP-GLC in the asymptotic regime when the number of users and files is kept constant while the packetization order is sent to infinity. Denoting by $E[R^{\mathrm{RAP}-\mathrm{GLC}}\left(P,Q,t\right)]$ the asymptotic average achievable rate of RAP-GLC for a fixed threshold *t*, the threshold parameter *t* is optimized to minimize $E[R^{\mathrm{RAP}-\mathrm{GLC}}\left(P,Q,t\right)]$. Hence, denoting by ${\bar{R}}^{\mathrm{RAP}-\mathrm{GLC}}$ the average rate achieved by RAP-GLC with optimized *t*, i.e.,
$${\bar{R}}^{\mathrm{RAP}-\mathrm{GLC}}=\min_{t}E\left[R^{\mathrm{RAP}-\mathrm{GLC}}\left(P,Q,t\right)\right],$$
we have that
$${\bar{R}}^{\mathrm{RAP}-\mathrm{GLC}}\leq \min\{E\left[R^{\mathrm{RAP}-\mathrm{GLC}}\left(P,Q,n\right)\right],E\left[R^{\mathrm{RAP}-\mathrm{GLC}}\left(P,Q,0\right)\right].$$

Since $E[R^{\mathrm{RAP}-\mathrm{GLC}}\left(P,Q,0\right)]$ is just the rate achieved via naive multicasting, then, an upper bound on the average asymptotic performance of RAP-GLC can be obtained by upper bounding $E[R^{\mathrm{RAP}-\mathrm{GLC}}\left(P,Q,n\right)]$ which can be obtained by generalizing the asymptotic performance analysis of RAP-GCC derived in [10,12] using conventional graph coloring in the homogeneous shared link caching network to the case of using local coloring in the heterogeneous caching network. Specifically, we extend the order-optimality analysis under single per-user requests (L=1) in the asymptotic regime of B→∞ [10,12], to that under multiple L>1 per-use requests [13,14]. These theoretical results will serve as rate lower bounds for the finite-length performance of our proposed algorithms.

Let $L=\max_{u}L_{u}$ and order $L_{u},u\in U$ as a decreasing sequence $L_{\left[1\right]}\geq L_{\left[2\right]}\geq L_{\left[3\right]},\cdots ,L_{\left[n\right]}$, where $L_{\left[i\right]}$ is the *i*-th largest $L_{u}$ and [i]=u for some u∈U. It can be seen that $L_{\left[1\right]}=\max_{u}L_{u}$ and $L_{\left[n\right]}=\min_{u}L_{u}$. Let $n_{j}=\sum_{\left[i\right]}1\left\{L_{\left[i\right]}-j\geq 0\right\}>0$, where $1\leq j\leq L_{\left[1\right]}$ and 1{·} is the indicator function. Let $U_{n_{j}}=\left\{\left[i\right]\in U:1\left\{L_{\left[i\right]}-j\geq 0\right\}\right\}$. In the next theorem, we provide a performance guarantee of the RAP-GLC algorithm.

**Theorem** **1.**
*For any given m, n, $M_{u}$, the random caching distribution P and the random request distribution Q, the average achievable rate of the RAP-GLC algorithm, ${\bar{R}}^{\mathrm{RAP}-\mathrm{GLC}}$ satisfies*
(12)$$\begin{array}{c} {\bar{R}}^{\mathrm{RAP}-\mathrm{GLC}}\leq \min\left\{\psi \left(P,Q\right),\bar{m}-\bar{M}\right\}, \end{array}$$
*when B→∞, where,*
(13)$$\begin{array}{cc} \bar{m} & =\sum_{f=1}^{m}\left(1-\prod_{u=1}^{n}{\left(1-q_{f,u}\right)}^{L_{u}}\right), \end{array}$$
(14)$$\begin{array}{cc} \bar{M} & =\sum_{f=1}^{m}\left(1-\prod_{u=1}^{n}{\left(1-q_{f,u}\right)}^{L_{u}}\right)\min_{u}p_{f,u}M_{u}, \end{array}$$
$$\begin{array}{cc} \psi (P,Q) & =\sum_{j=1}^{L}\sum_{\ell =1}^{n}\sum_{U^{\ell }\subset U_{n_{j}}}\sum_{f=1}^{m}\sum_{u\in U^{\ell }}\rho_{f,u,U^{\ell }}\lambda \left(u,f,U^{\ell }\right), \end{array}$$
*with $U^{\ell }$ denoting a set of users with cardinality ℓ,*
(15)$$\begin{array}{c} \lambda \left(u,f_{u},U^{\ell }\right)=\left(1-p_{f_{u},u}M_{u}\right)\times \prod_{k\in U^{\ell }\backslash \left\{u\right\}}\;\;\left(p_{f_{u},k}M_{k}\right)\;\;\prod_{k\in U\backslash U^{\ell }}\;\;\left(1-p_{f_{u},k}M_{k}\right) \end{array}$$
*and*
$$\begin{array}{c} \rho_{f,u,U^{\ell }}\overset{\Delta }{=}P\left(f=\arg\;\max_{f_{u}\in f\left(U^{\ell }\right)}\;\;\;\lambda \left(u,f_{u},U^{\ell }\right)\right), \end{array}$$
*denoting the probability that f is the file whose $p_{f,u}$ maximizes the term $\lambda (u,f_{u},U^{\ell })$ among $f(U^{\ell })$ (the set of files requested by $U^{\ell }$).*


**Proof.** See Appendix A. □

Using the explicit expression for ${\bar{R}}^{\mathrm{RAP}-\mathrm{GLC}}$ in Theorem 1, we can optimize the caching distribution for a wide class of heterogeneous network models to minimize the number of transmissions. We use $P^{*}$ to denote the caching distribution that minimizes $R^{\mathrm{RAP}-\mathrm{GLC}}$.

**Remark** **1.**
*For the sake of the numerical evaluation of ψ(q,p), it is worthwhile to note that the probabilities $\rho_{f,u,U^{\ell }}$ can be easily computed as follows. Given the subset of users, $U^{\ell }$ of cardinality ℓ, let $J_{u_{1}},\ldots ,J_{u_{\ell }}$ denote ℓ i.i.d. random variables each of them distributed over F with pmf $q_{u_{i}}$, with i=1,…,ℓ. Since $\lambda \left(u_{1},J_{u_{1}}U^{\ell }\right),\cdots ,\lambda \left(u_{\ell },J_{u_{\ell }},U^{\ell }\right)$ are i.i.d., the CDF of $Y_{\ell }\overset{\Delta }{=}\max\left\{\lambda \left(u_{1},J_{u_{1}},U^{\ell }\right),\cdots ,\lambda \left(u_{\ell },J_{u_{\ell }},U^{\ell }\right)\right\}$ is given by*
(16)$$\begin{array}{cc} P\left(Y_{\ell }\leq y\right) & =\prod_{i=1}^{\ell }P\left(\lambda (u_{i},J_{u_{i}},U^{\ell })\leq y\right)\\ & =\prod_{i=1}^{\ell }\left(\sum_{j\in F:\lambda (u_{i},j,U^{\ell })\leq y}q_{u_{i},j}\right). \end{array}$$
*Hence, it follows that*
(17)$$\begin{array}{cc} \rho_{f,u,U^{\ell }} & =P(Y_{\ell }=\lambda \left(u,f,U^{\ell }\right))\\ & =\prod_{i=1}^{\ell }\left(\sum_{j\in F:g_{\ell }\left(j\right)\leq \lambda \left(u,f,U^{\ell }\right)}q_{u_{i},j}\right)-\prod_{i=1}^{\ell }\left(\sum_{j\in F:g_{\ell }\left(j\right)<\lambda \left(u,f,U^{\ell }\right)}q_{u_{i},j}\right), \end{array}$$
*which can be easily computed by sorting the values $\left\{\lambda \left(u_{i},j,U^{\ell }\right):j\in F,u_{i}\in U^{\ell }\right\}$.*


Nevertheless, as shown in [21], when *B* is finite or is not exponential in *n*, the performance of RAP-GLC can degrade significantly, compromising the promising multiplicative caching gain, although it is already an improved version of RAP-GCC in [12]. This brings us to the other main contribution where we propose a new algorithm that preserves the gain due to coded multicasting even when *B* is finite.

### 4.2. Randomized Aggregate Popularity-Hierarchical Greedy Local Coloring (RAP-HgLC) for Finite-Length Packetization

Similarly to RAP-GLC, RAP-HgLC has a predetermined parameter t∈{0,⋯,n} that is optimized to minimize its associated average achievable rate. However, in the RAP-HgLC algorithm, we arrange the vertices in a hierarchy and use this to design a more careful coloring algorithm. The key idea of RAP-HgLC is to exploit the labeling of each vertex more efficiently. More specifically, as in RAP-GLC, RAP-HgLC associates to each vertex *v* a label or tag, composed by the two fields $K_{v}\equiv \left(T_{D}\left(v\right),T_{C}\left(v\right)\right)$, defined in (Equation 10) and (Equation 11).

#### 4.2.1. RAP-HgLC Algorithm Description

Before jumping into the algorithm, we introduce the following useful notations and their definitions.
$G_{i}$: The *i*-th layer, $G_{i}$ is initialized with the set of vertices $\left\{v:|T_{v}|=i\right\}$ and at any point in the algorithm contains only vertices with $|K_{v}|\geq i$. $G_{i}$ is updated continuously in the algorithm. Therefore, higher numbered layers contain vertices with greater popularity.$W_{1}\subset G_{i}$: a subset of $G_{i}$ consists of all the vertices with $|K_{v}|=i$ as well as a certain number of vertices with higher popularity (if available at any iteration), defined as
(18)$$\begin{array}{c} W_{1}=\left\{v\in G_{i}:\min_{v\in G_{i}}|K_{v}\left|\leq \right|K_{v}|\leq \min_{v\in G_{i}}\left|K_{v}\right|\right.\left.+\left\lfloor a\left(\max_{v\in G_{i}}|K_{v}|-\min_{v\in G_{i}}\left|K_{v}\right|\right)\right\rfloor \right\}, \end{array}$$
where a∈[0,1] is a design parameter and $W_{1}$ is updated with every iteration.$Q_{i}$ (see Algorithm 2): another subset of $G_{i}$ that is updated every iteration.$W_{2}\subset Q_{i}$: a subset of vertices in $Q_{i}$ defined as:
(19)$$\begin{array}{c} W_{2}=\left\{v^{\prime }\in Q_{i}:\min_{v^{\prime }\in Q_{i}}|K_{v^{\prime }}\left|\leq \right|K_{v^{\prime }}|\leq \min_{v^{\prime }\in Q_{i}}\left|K_{v^{\prime }}\right|\right.+\left.\left\lfloor b\left(\max_{v^{\prime }\in Q_{i}}|K_{v^{\prime }}|-\min_{v^{\prime }\in Q_{i}}\left|K_{v^{\prime }}\right|\right)\right\rfloor \right\}, \end{array}$$
where b∈[0,1] is another design parameter.

Based on the above definitions, it follows that the total set vertices V forms an *n*-layer hierarchy with the *i*-th layer composed of the set of vertices $G_{i}$.

**Key Idea:** Starting from layer *n*, at any layer i≤n, the RAP-HgLC algorithm attempts to form an independent set of size at least *i*; when there are no more such independent sets, all remaining packets are dropped to layer i−1, and transmission actions on those packets are *deferred* to later layers. This is the key difference between RAP-HgLC and RAP-GLC. That is, RAP-HgLC makes an extra effort to place nodes with large labels into large independent sets.

We will now describe how the above key idea is implemented in RAP-HgLC. The  RAP-HgLC algorithm forms large independent sets in a “top-down” fashion, starting with the highest layer, and iteratively moving to lower layers until layer 1. The following two steps are performed at each layer:**Step I**: The first step is similar to that in RAP-GLC algorithm. Given a vertex *v*, the algorithm first checks if the cardinality of $T_{D}\left(v\right)$ is higher than *t*, i.e., $|T_{D}\left(v\right)|>t$ then all the vertices $v^{\prime }$ such that $\rho \left(v\right)=\rho \left(v^{\prime }\right)$ are colored with the same color. If $|T_{D}\left(v\right)|\leq t$ then the algorithm greedily finds independent sets of size *i*, where every vertex *v* in the independent set (Algorithm 2, Line 20) has the same $K_{v}$ (Algorithm 2, Line 19). After removing these vertices, the rest of the vertices in $G_{i}$ are left for the second step.**Step II**: A candidate pool of vertices $W_{1}\subseteq G_{i}$ is created. This set contains vertices *v* such that $|K_{v}|$ being close to the smallest available $|K_{v}|$’s. We randomly pick a vertex *v* from $W_{1}$ (Algorithm 2, Line 31). The design parameter *a* determines how close is the picked $|K_{v}|$ to the smallest available ones. We gradually form an independent set of size *i* with *v* included as follows: Form another set $W_{2}$ (Algorithm 2, Line 34), excluding *v*, whose vertices have $|K_{v^{\prime }}|$ that is bigger but closer to that of *v* determined by *b*, sample repeatedly with replacement from it to grow the independent set. If an independent set of size at least *i* cannot be formed, we drop the vertex *v* to the lower layer $G_{i-1}$, and take it into account in the next layer iteration. Otherwise, we assign a color to the independent set. $W_{1}$ is repeatedly formed and random sampling from $W_{1}$ repeated till every vertex in $G_{i}$ is dropped or colored.

**Algorithm 2** HgLC1: C=∅;2: c=∅; 3: choose a∈[0,1] 4: choose b∈[0,1] 5: **for all**
i=n,n−1,⋯,2,1
**do** 6:  **for all**
v∈Gi and |Kv|=i
**do** 7:   I={v}; 8:   Let V′=V\{v}; 9:   **if** { |TD(v)|>t } **then**10:    **for all**
v′∈V′ with ρ(v′)=ρ(v)
**do**11:     I=I∪v′;12:    **end for**13:    Color all the vertices in I by c∉C;14:    Let c[I]=c;15:    **for all**
i=n,n−1,⋯,2,1
**do**16:     Gi=Gi\I;17:    **end for**18:   **else**19:    **for all**
v′∈Gi\I with Kv′≡Kv
**do**20:     **if** {There is no edge between v′ and I} **then**21:      I=I∪v′;22:     **end if**23:    **end for**24:    **if**
|I|=i
**then**25:     Color all the vertices in I by c∉C;26:     c[I]=c, C=C∪c;27:     Gi=Gi\I;28:    **end if**29:   **end if**30:  **end for**31:  **for all**
v∈Gi with *v* randomly picked from W1⊂Gi
**do**32:   I={v};33:   Qi=Gi\I;34:   **for all**
v′∈Qi with v′ randomly picked from W2⊂Qi. **do**35:    **if** {Kv′⊃Kv } ∩ {No edge between v′ and I} **then**36:     I=I∪v′;37:     Qi=Qi\{v′};38:    **else**39:     Qi=Qi\{v′};40:    **end if**41:   **end for**42:   **if**
|I|≥i
**then**43:    Color all the vertices in I by c∉C;44:    c[I]=c, C=C∪c;45:    Gi=Gi\I;46:   **else**47:    Gi=Gi\{v}, Gi−1=Gi−1∪{v};48:   **end if**49:  **end for**50: **end for**51: c= LocalSearch(HC,W,c,C);52: **return** the local coloring number maxv∈V|c(N+(v))| and the corresponding color assignment c(N+(v)) for each *v*;

**Remark** **2.**
*Please note that RAP-GLC goes through the same Step I as RAP-HgLC, and then simply assigns a different color to each remaining uncolored vertex. On the other hand, Step II in RAP-HgLC tries to find further independent sets among the remaining uncolored vertices. It is this extra step that guarantees the performance of RAP-HgLC to be no worse than that of RAP-GLC.*


The RAP-HgLC algorithm, when operating on the *i*-th layer, *always* colors at least *i* vertices with the same color. Please note that if there are remaining vertices when reaching layer 1, all such vertices will be colored, each with a different color.

To further reduce the required number of colors, we use a function called LocalSearch (Algorithm 2, Line 51), which is described in Algorithm 3. It works in an iterative fashion by replacing the current solution with a better one if there exists. It terminates when no better solutions can be found. In particular, the local search algorithm has the purpose of checking the redundancy of each color c∈C, to eventually decrease the current objective function value |C|. In more detail, the local search computes, iteratively for each color c∈C, the set $J_{c}$ of all vertices colored with color *c*, and performs the following steps:For each vertex $i\in J_{c}$, if there is a color $c^{\prime }\in C$, $c^{\prime }\neq c$ that is not assigned to any adjacent vertex j∈Adj(i), then assign vertex *i* with color $c^{\prime }$;Color *c* is removed from the set C if and only if in the previous step it has been possible to replace *c* with some color $c^{\prime }\neq c$ for all vertices in $J_{c}$.

Finally, in Algorithm 2, Line 52, we compute the local coloring number.

**Algorithm 3** LocalSearch($H_{C,W},c,C$) 1: **for all**
c∈C
**do** 2:  Let $J_{c}$ be the set of vertices whose color is *c*; 3:  Let B=∅; 4:  Let $\hat{c}=c$; 5:  **for all**
$i\in J_{c}$
**do** 6:   A=∅; 7:   **for all**
j∈N(i)
**do** 8:    A=A∪c[j]; 9:    **if**
C\A≠∅
**then**10:     c′ is chosen uniformly at random from C\A;11:     c^[i]=c′;12:     B=B∪{i};13:    **end if**14:   **end for**15:   **if**
|B|=|Jc|
**then**16:    c=c^;17:    C=C\c;18:   **end if**19:  **end for**20: **end for**21: **return**
c;

To illustrate the RAP-HgLC algorithm, we present the following example.

**Example** **3.**
*Consider a shared link network with n=3 users: U={1,2,3}, and m=3 files: F={A,B,C}. Each file is partitioned into *4* packets. For example, $A=\{A_{1},A_{2},A_{3},A_{4}\}$. For the caching part, let user *1* cache $\left\{A_{1},B_{1},B_{2},B_{3},B_{4},C_{2}\right\}$, user *2* cache $\left\{A_{1},A_{2},A_{3},A_{4},B_{1},C_{2},C_{3}\right\}$, user *3* cache $\left\{A_{1},A_{2},A_{3},C_{1},B_{2},B_{3}\right\}$. Then, let users {1,2,3} request files {A,B,C} respectively. Equivalently, user *1* requests $A_{2},A_{3},A_{4}$; user *2* requests $B_{2},B_{3},B_{4}$; user *3* requests $C_{2},C_{3},C_{4}$. Then, we have $K_{A_{2}}=\left\{1,2\;\;3\right\}$; $K_{A_{3}}=\left\{1,2\;\;3\right\}$; $K_{A_{4}}=\left\{1,2\right\}$; $K_{B_{2}}=\left\{2,1\;\;3\right\}$; $K_{B_{2}}=\left\{2,1\;\;3\right\}$; $K_{B_{4}}=\left\{2,1\right\}$; $K_{C_{2}}=\left\{3,1\;\;2\right\}$; $K_{C_{3}}=\left\{3,2\right\}$; $K_{C_{4}}=\left\{3\right\}$ (here $C_{4}$ is requested by user *3* and not cached anywhere).*

*The RAP-HgLC algorithm works as follows. For i=n=3, $G_{3}=\left\{A_{2},A_{3},B_{2},B_{3},C_{2}\right\}$, let $v=A_{2}$, then it can be found that $B_{2}$ and $C_{2}$ would be in I, hence $I=\{A_{2},B_{2},C_{2}\}$. Now since |I|=n=3, we color $A_{2},B_{2},C_{2}$ by black (see Figure 4). Then $G_{i}=G_{i}\backslash I=\left\{A_{3},B_{3}\right\}$. In the following loop, since we cannot find a set I with |I|=n=3, we move to Line 19. Then since we cannot find a I with |I|≥n=3, then we do $G_{2}=G_{2}\cup \left\{A_{3}\right\}$, and then $G_{2}=G_{2}\cup \left\{B_{3}\right\}$. Therefore, we obtain $G_{2}=\left\{A_{3},A_{4},B_{3},B_{4},C_{3}\right\}$. Now we go to Line 5 (start next loop). For i=n−1=2, in this loop, we first pick $v=A_{4}$, then we can find $I=\{A_{4},B_{4}\}$. We color $\left\{A_{4},B_{4}\right\}$ by blue (see Figure 4). Now $G_{2}=G_{2}\backslash \left\{A_{4},B_{4}\right\}=\left\{A_{3},B_{3},C_{3}\right\}$. Then in Line 19, we find the vertex with smallest length of $K_{v}$ (let a=0), which is $C_{3}$ with $K_{C_{3}}=\left\{3,2\right\}$, then we have $I=\{C_{3}\}$ and $Q_{2}=\left\{A_{3},B_{3}\right\}$, then in the next loop, we can find $I=\{C_{3},B_{3}\}$. We color $I=\{C_{3},B_{3}\}$ by red (see Figure 4). Now $G_{2}=G_{2}\backslash \left\{C_{3},B_{3}\right\}=\left\{A_{3}\right\}$. Since there is no I with |I|≥2, then we do $G_{1}=G_{1}\cup \left\{A_{3}\right\}=\left\{C_{4},A_{3}\right\}$. Then we go to next loop i=n−2=1. Then we can see that $I=\{C_{4}\}$, and we color $\left\{C_{4}\right\}$ by purple (see Figure 4). Then $G_{1}=G_{1}\backslash \left\{C_{4}\right\}=\left\{A_{3}\right\}$. Hence, we can find $I=\{A_{3}\}$ and we color $\left\{A_{3}\right\}$ by brown.*

*According to Figure 4, the total number of required colors is 5, while the maximum number of colors required locally by each user is 4. For the naive multicasting, since it only allows the vertices represented the same packet to be colored by the same color, the total number of required colors is 9. The corresponding rate is given by 9/4. Hence, the final rate achieved by RAP-HgLC with local coloring is no more than min{4/4,9/4}=1. For the interested reader, it can be verified that if the GCC algorithm, designed for B→∞, as proposed in [10], is used, the corresponding number of required colors is *6*.*


The complexity of RAP-HgLC can be computed as follows. For the hierarchical coloring procedure (Line 5–50 in Algorithm 2), the complexity is $O(n|V{|}^{2})$, and the complexity of local search procedure is O(|E|). Therefore, the running time complexity of RAP-HgLC is given by ${O(n|V|}^{2}+\left|E|\right)={O(n|V|}^{2})$. Since |V|≤nB, the running time complexity of RAP-HgLC is $O\left(n^{3}B^{2}\right)$.

#### 4.2.2. RAP-HgLC Performance Analysis

For the general heterogeneous network setting, tight upper bounds on the asymptotic (B→∞) average achievable rate of RAP-HgLC are quite complex to derive, even though a simple (but not necessarily tight) upper bound on the asymptotic performance can be obtained considering the asymptotic average rate of RAP-GLC (see Remark 2).

Regarding the finite-length regime, in [21] we derived a tight upper bound on the performance of RAP-HgLC for the simpler case of homogenous networks under worst-case demands. Specifically, the bound in [21], requires *B* to be $\tilde{O}\left({\left(\frac{m}{M}\right)}^{g+2}\right)$ (where $\tilde{O}$ hides some poly log terms) to achieve a worst-case rate of at most $\frac{n}{g}$. This approximately matches a lower bound of $\tilde{O}\left({\left(\frac{m}{M}\right)}^{g}\right)$ derived in the same work for any coloring algorithm, showing, for the simpler homogenous network setting, the optimality of RAP-HgLC among all graph-coloring-based algorithms.

For the more complex setting where demands arise from popularity distributions and every user requests multiple files, the finite-length performance of RAP-HgLC is investigated in Section 6 via numerical analysis, where we show how the RAP-HgLC is able to recover most of the multiplicative caching gain even with very moderate packetization order.

## 5. Tradeoff between Number of Requests and Code Length

As mentioned earlier, in the simpler homogenous scenario, the authors in [21] showed that under worst-case demands, to achieve a gain *g* over conventional naive multicasting, it is necessary for *B* to grow exponentially with *g*. Intuitively, this is because a sufficiently large *B* is needed to create coded multicast transmissions that are useful for multiple users. However, when each user makes multiple requests, the number of requests $L_{u}=L$ can play a similar role to that of *B*, such that the requirement for *B*, and hence the resulting computational complexity can be reduced. For ease of analysis, in this section, we assume that all users place the same number of requests ($L_{u}=L$).

In the following, under either worst-case or uniform demands, we show the sufficient conditions on *B* and *L* that guarantee achieving a gain $g=\frac{Mn}{m}$. From this result, we can obtain the regime where *B* and *L* are interchangeable (*L* plays an equivalent role to *B*). Note that it can be shown that the number of file transmissions under both worst-case and uniform demands have the same order.

We consider two cases for the range of *B*: the case of B=1, and the case of $B=\omega \left(\frac{m}{M}\right)$. The regime where $1<B=O\left(\frac{m}{M}\right)$ is out of the scope of this paper.

When B=1, the cache placement algorithm becomes scalar uniform cache placement (SUP), in which each user caches *M* entire files chosen uniformly at random. For simplicity, we let *M* be a positive integer. Then, as shown in [14], letting L→∞ as a function of n,m,M, we obtain the following theorem.

**Theorem** **2.**
*When B=1 and M=ω(1), for the shared link caching network with n users, library size m, storage capacity M, and L distinct per-user requests (nL≤m), if (i) $\frac{M}{m}\leq \frac{1}{2}$ and*
(20)$$\begin{array}{cc} L & =\omega \left(\max\left\{\frac{nM}{m}{\left(\frac{m}{M}\right)}^{n}\frac{1}{\left(1-\frac{M}{m}\right)},\left(\frac{{\left(nM\right)}^{\frac{1}{2(1-\varepsilon )}}}{\left(\frac{m}{M}-1\right)\left(1-{\left(1-\frac{M}{m}\right)}^{n}\right)}\right)\right\}\right), \end{array}$$
*or (ii) $\frac{M}{m}\geq \frac{e}{1+e}$ and*
(21)$$\begin{array}{cc} L & =\omega \left(\max\left\{{\left(\frac{m}{m-M}\right)}^{n},\left(\frac{{\left(nM\right)}^{\frac{1}{2(1-\varepsilon )}}}{\left(\frac{m}{M}-1\right)\left(1-{\left(1-\frac{M}{m}\right)}^{n}\right)}\right)\right\}\right), \end{array}$$
*where ε is an arbitrarily small number,*

*then, the achievable rate of RAP-GLC is upper bounded by*
$$\begin{array}{cc} & \lim_{n,m\to \infty }P\left(R^{\mathrm{SUP}-\mathrm{GLC}}\leq \left(1+o\left(1\right)\right)\min\left\{L\left(\frac{m}{M}-1\right),Ln,m-M\right\}\right)=1. \end{array}$$


**Proof.** See Appendix B. □

From Theorem 2, we can see that when *L* and *M* are large enough, instead of requiring a large *B* and packet-level coding, a simpler file-level coding scheme is sufficient to achieve the same order-optimal rate. We remark, however, that the range of the parameter regimes in which this result holds is limited due to the requirement of a large *M* and *L*. Next, we focus on another parameter regime, when $B=\omega \left(\frac{m}{M}\right)$, and find the achievable tradeoff between *B* and *L*.

**Theorem** **3.**
*When $B=\omega \left(\frac{m}{M}\right)$, for the shared link caching network with n users, library size m, storage capacity M, and L distinct per-user requests (nL≤m), if (i) $\frac{M}{m}\leq \frac{1}{2}$, and*
(22)$$\begin{array}{cc} B & =\omega \left(\max\left\{\frac{nM}{Lm}{\left(\frac{m}{M}\right)}^{n}\frac{1}{\left(1-\frac{M}{m}\right)},\frac{{\left(nM\right)}^{\frac{1}{2(1-\varepsilon )}}}{L\left(\frac{m}{M}-1\right)\left(1-{\left(1-\frac{M}{m}\right)}^{n}\right)}\right\}\right), \end{array}$$
*or (ii) $\frac{M}{m}\geq \frac{e}{1+e}$, and*
(23)$$\begin{array}{cc} B & =\omega \left(\max\left\{\frac{1}{L}{\left(\frac{m}{m-M}\right)}^{n},\frac{{\left(nM\right)}^{\frac{1}{2(1-\varepsilon )}}}{L\left(\frac{m}{M}-1\right)\left(1-{\left(1-\frac{M}{m}\right)}^{n}\right)}\right\}\right), \end{array}$$
*where ε is an arbitrarily small number,*

*Then, the achievable rate of RAP-GLC is upper bounded by*
(24)$$\begin{array}{c} \lim_{n,m\to \infty }P\left(R^{\mathrm{SUP}-\mathrm{GLC}}\leq \left(1+o\left(1\right)\right)\min\left\{L\left(\frac{m}{M}-1\right),Ln,m-M\right\}\right)=1. \end{array}$$


**Proof.** See Appendix C. □

If we particularize Theorem 1 to the homogenous network setting under uniform demands, we see that the rate achieved by RAP-GLC is upper bounded by the same expression given in (24). Hence, from Theorem 2, we can see that when *L* is large enough, instead of requiring a very large *B*, an intermediate value of $B=\omega \left(\frac{m}{M}\right)$ is sufficient to achieve the same order-optimal rate. In practice, it is important to find the right balance and tradeoff between *B* and *L* given the remaining system parameters. In Section 6, we show via simulation that a similar tradeoff holds also for RAP-HgLC.

## 6. Simulations and Discussions

In this section, we numerically evaluate the performance of the two polynomial-time algorithms described in Section 4, RAP-GLC and RAP-HgLC, in the finite-length regime characterized by the number of packets per file *B*.

Recall that the caching distribution $P^{*}$ is to be optimized to minimize the number of transmissions. Since the distribution $P^{*}$ resulting from minimizing the right-hand side of (Equation 12) may not admit an analytically tractable expression in general, in the following numerical results, we restrict the caching distribution to take the form of a *truncated uniform distribution*
${\tilde{p}}_{u}$, as described in [12]:(25)$$\begin{array}{c} {\tilde{p}}_{f,u}={\frac{1}{\tilde{m}}}_{u},\;f\leq {\tilde{m}}_{u}\\ {\tilde{p}}_{f,u}=0,\;f\geq \tilde{m}+1 \end{array}$$
where the cut-off index ${\tilde{m}}_{u}\geq M$ is a function of the system parameters that is optimized to minimize the right-hand side of (Equation 12). The intuition behind the form of ${\tilde{p}}_{u}$ in (Equation 25) is that each user caches the same fraction of (randomly selected) packets from each of the most ${\tilde{m}}_{u}$ popular files, and does not cache any packet from the remaining $m-{\tilde{m}}_{u}$ least popular files. We point out that when ${\tilde{m}}_{u}=M$, this cache placement coincides with the LFU (Least Frequently Used) caching policy. Thus, this cache placement is referred to as *Random LFU* (RLFU) [12], and the corresponding caching algorithms as RLFU-GLC and RLFU-HgLC. Recall that LFU discards the least frequently requested file upon the arrival of a new file to a full cache of size $M_{u}$ files. In the long run, this is equivalent to caching the $M_{u}$ most popular files [69].

In Figure 5 and Figure 6, we plot the average achievable rate, i.e., the average number of transmissions (normalized by the file size) as a function of the cache size for RLFU-GLC and RLFU-HgLC. For comparison, we also simulate the following algorithms:LFU, which has been shown to be optimal in single cache networks;RLFU-GLC with infinite file packetization (B→∞), whose performance guarantee is given in Theorem 1, and it is shown to be order optimal.

Regarding the LFU algorithm, the average achievable rate is given by
(26)$$E\left[R^{\mathrm{LFU}}\right]=\sum_{f=\min_{u}\left\{M_{u}\right\}+1}^{m}\left(1-\prod_{u\in U_{\left\{M_{u}<f\right\}}}{\left(1-q_{f,u}\right)}^{L_{u}}\right),$$
where $U_{\left\{M_{u}<f\right\}}$ denotes the set of users with $M_{u}<f$.

For simplicity, and to better illustrate the effectiveness of the proposed algorithms, especially under multiple per-user requests, we consider a scenario in which all users request files according to a Zipf demand distribution with parameter γ∈{0.2,0,4,0.5}, and all caches have size *M* files. Under Zipf demands, file *f* is requested with probability $\frac{f^{-\gamma }}{\sum_{i=1}^{m}i^{-\gamma }}$.

We consider two types of users. In Figure 5a and Figure 6a, users represent end devices requesting only one file each (L=1); while in Figure 5b and Figure 6b, they represent helpers/small-cells, each serving 10 end user devices, and consequently collecting L=10 requests.

In Figure 5a,b, we fix the total number of users *n* and the product between *L* and *B* (L×B=1000). Figure 5a plots the average rate for a network with n=40 users, γ=0.5, L=1, and B=1000. It is immediate to observe the impact of finite packetization on the multiplicative caching gain. In fact, as predicted by the theory (see [21]), the significant caching gain (with respect to LFU) quantified by the asymptotic performance of RAP-GLC (GLC with B=∞) is completely lost when using RAP-GLC with finite packetization (GLC with B=1000). On the other hand, RAP-HgLC remarkably preserves, at the expense of a slight increase in computational complexity, most of the multiplicative caching gain for the same value of file packetization. For example, in Figure 5a, if *M* doubles from M=200 to M=400, then the rate achieved by RAP-HgLC reduces from 15 to 5.7. Furthermore, RAP-HgLC can achieve a factor of 3.5 rate reduction from LFU for M=500. For the same regime, it is straightforward to verify that neither RAP-GLC nor LFU exhibit this property. Note from Figure 5a that to guarantee a rate of 10, RAP-GLC requires a cache size of M=500, while RAP-HgLC can reduce the cache size requirement to M=250, a 2× cache size reduction. Furthermore, while LFU can only provide an additive caching gain, additive and multiplicative gains may show indistinguishable when *M* is comparable to the library size *m*. Hence, one needs to pick a reasonably small *M* ($\frac{m}{n}<M\ll m$) to observe the multiplicative caching gain of RAP-HgLC.

Figure 5b shows the average rate for a network with n=40 helpers/small-cells, each serving 10 users making requests according to a Zip distribution with γ=0.5. Hence, the total number of distinct requests per helper is up to $L_{u}=10,\forall u\in \left\{1,\ldots 20\right\}$. In this case, we assume B=100 (instead of B=1000 in Figure 5a). In order to make easier the comparison with Figure 5a, we normalize the achievable rate (number of transmissions) by the file size and the number of requests.

Note from Figure 5a,b that as predicted by Theorem 3, when $L_{u}$ increases (from $L_{u}=1$ to $L_{u}=10$), almost the same multiplicative caching gain can be achieved with a smaller *B* (from B=1000 to B=100). In fact, from Figure 5a,b, we see that under RAP-HgLC, the average rate per request for B=100 and L=10 is almost the same as the average rate per request for B=1000 and L=1. This confirms the interesting tradeoff between *B* and *L* established in Theorem 3.

We can observe a similar behavior in Figure 6a,b. Figure 6a plots the average rate for a network with n=80 users, γ=0.4, L=1, and B=200. RAP-HgLC is able to preserve most of the multiplicative caching gain for the same values of file packetization. For example, in Figure 6a, if *M* doubles from M=200 to M=400, then the rate achieved by RAP-HgLC essentially halves from 20 to 10. Furthermore, RAP-HgLC can achieve a factor of 5 rate reduction from LFU for M=500. Note from Figure 6a that to guarantee a rate of 20, RAP-GLC requires a cache size of M=500, while RAP-HgLC can reduce the cache size requirement to M=200, a 2.5× cache size reduction.

Figure 6b plots the average rate for a network with n=20 helpers/small-cells, each serving 10 users making requests according to a Zip distribution with γ=0.2. Hence, the total number of distinct requests per helper is up to $L_{u}=10,\forall u\in \left\{1,\ldots 20\right\}$. In this case, we assume B=100. Differently from Figure 5b, here we plot the average rate without normalizing it by the number of requests.

Note from Figure 6a,b that, as predicted by Theorem 3, when $L_{u}$ increases (from $L_{u}=1$ to $L_{u}=10$), almost the same multiplicative caching gain can be achieved with a smaller *B* (from B=200 to B=100). In fact, from Figure 6a,b, we see that under RAP-HgLC, the average rate per request for B=100 and L=10 is almost the same as the average rate per request for B=200 and L=1. For example, for M=200, B=100, and L=10, the per request average rate achieved by RAP-HgLC is 0.3, while for M=200 and B=200, is 0.25. This again confirms the tradeoff between *B* and *L* stated in Theorem 3.

Furthermore, from Figure 5 and Figure 6, we notice that increasing the Zip parameter reduces the gains with respect to LFU. This is explained by the fact that when aggregating multiple requests, there is a higher number of overlapping requests, which increases the opportunities for naive multicasting (as clearly characterized in [13]). Note, however, that RAP-HgLC can remarkably keep similar gains with respect to LFU in this multiple-request setting, and approach the asymptotic performance even with just B=100 packets per file, confirming the effectiveness of the local graph coloring and extra processing procedures in RAP-HgLC.

## 7. Conclusions

Coded multicasting has been shown to be a promising approach to significantly reduce the traffic load in wireless caching networks. However, most existing schemes require the number of packets per file to grow exponentially with the number of users. To address this challenge, in this paper we focused on a heterogeneous shared link caching network model and designed novel coded multicast algorithms based on local graph coloring that exhibit polynomial-time complexity in all the system parameters, and preserve the asymptotically proven multiplicative caching gain for finite file packetization. We also demonstrated that the number of packets per file can be traded-off with the number of requests collected by each cache, such that the same multiplicative caching gain can be preserved. Simulation results confirm the superiority of the proposed schemes and illustrate the tradeoff between request aggregation and computational complexity (driven by the packetization order), shedding light into the practical achievability of the promising multiplicative caching gain in next generation wireless networks. 

## Figures and Tables

**Figure 1 entropy-21-00324-f001:**
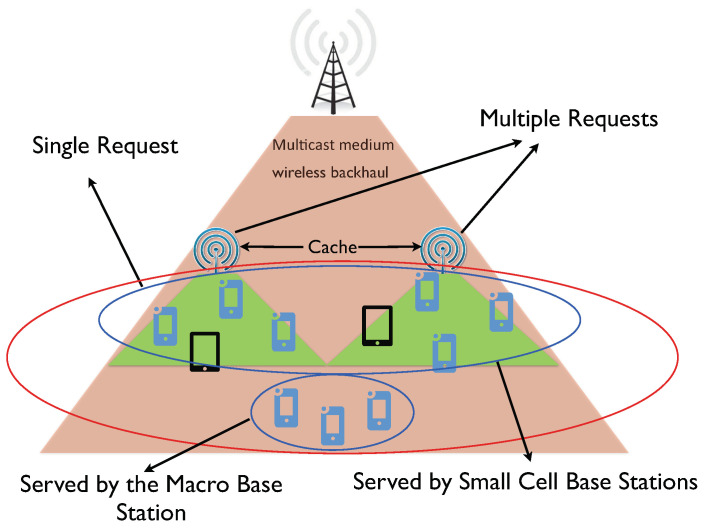
An example of the network model, which consists of a source node (base station in this figure) with access to the content library and connected to the users via a shared (multicast) link. Each user (end users and small-cell base stations) may have different cache size and request a different number of files according to their own demand distribution.

**Figure 2 entropy-21-00324-f002:**
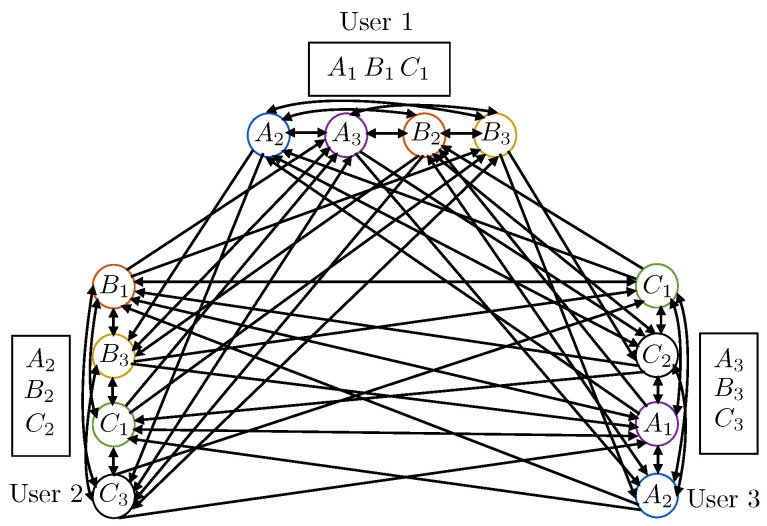
An example of the construction of the directed conflict graph (this figure needs to be viewed in color). The color of each circle in this figure represents the coloring of each vertex.

**Figure 3 entropy-21-00324-f003:**
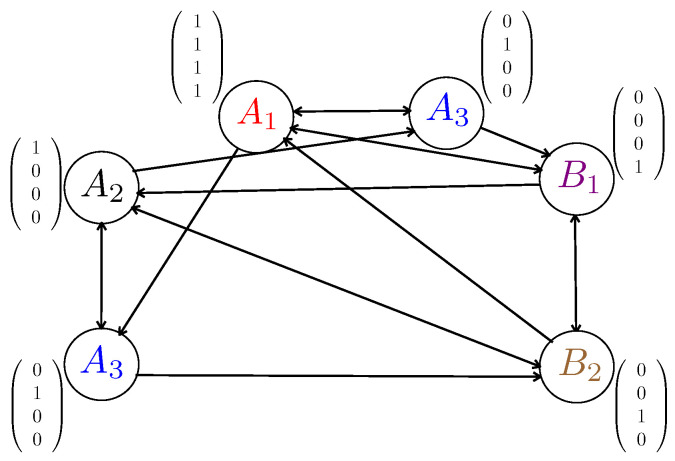
An illustration of coded multicast codewords construction based on local coloring (this figure needs to be viewed in color). The total number of colors is ξ=5, and the local coloring number is $\xi_{\mathrm{lc}}=4$.

**Figure 4 entropy-21-00324-f004:**
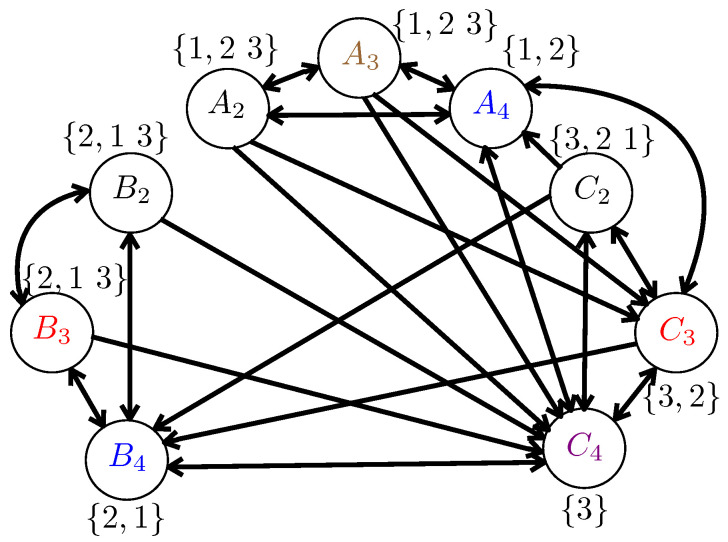
One example for the RAP-HgLC algorithm (this figure needs to be viewed in color).

**Figure 5 entropy-21-00324-f005:**
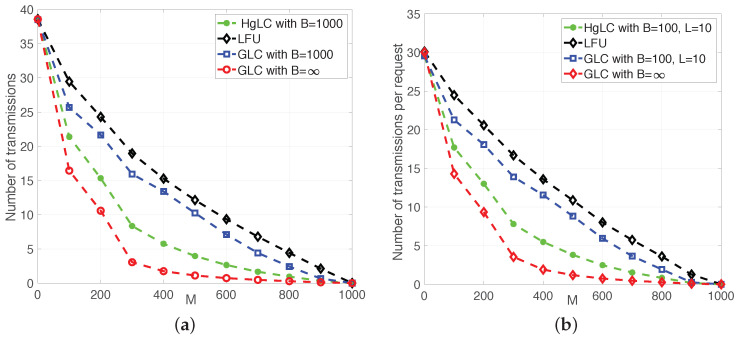
Average number of transmissions in a heterogeneous shared link caching network with m=1000. (**a**) n=40, L=1, γ=0.5; (**b**) n=40, L=10, γ=0.5.

**Figure 6 entropy-21-00324-f006:**
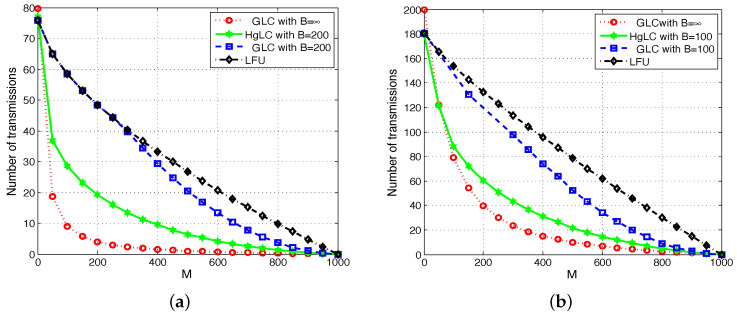
Average number of transmissions in a heterogeneous shared link caching network with m=1000. (**a**) n=80, L=1, γ=0.4; (**b**) n=20, L=10, γ=0.2.

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
