# Peer review of "Efficient Algorithms for Coded Multicasting in Heterogeneous Caching Networks"

_entropy, 2019, doi:10.3390/e21030324_

Round 1

Reviewer 1 Report

The paper described a coded multicasting algorithm for caching networks.

In particular, the authors consider the scenario of caching with random cache placement in which caches of different sizes collect multiple requests according to different demand distributions.

The paper is well written and has extensive connection to the available literature and research directions: this clearly results from the extensive, combined, contributions of the authors to this research direction, see [1,2,5,12,14,15,18,19,20,21…] so that it would be quicker to list the papers in the bibliography that do not contain at least one of the authors than otherwise.

Indeed, this is somewhat the limit of the paper: the index coding problem presented here arises from the layering and combining of scenarios so that it has somewhat lost its immediacy and ease of understanding.

For instance, understanding the example in Fig. 2 required me quite some time, given the sheer number of arrows in the figure. Although I understand that this is the correct representation of this example, it is indicative that this is the minimal example that can be produced to present the problem setting.I would appreciate if the author would try to further improve the clarity of the problem presentation and the relevance of the different aspects of the problem.

The paper contribution are incremental and build upon the results in [12,14] and result in a multicast scheme that is optimal in scaling. Such results are interesting and timely, as they show that asymptotic multiplicative caching gain are attainable even for finite file packetization.

The complexity considerations are also interesting and relevant to practical applications.

Personally, I have the impression that the simulation section is somewhat rushed and incomplete. Although proposed coding schemes are motivated in the information theoretical settings, they are yet distilled in two algorithms that can be analyzed in precise detail. I think that this section can be enriched by investigating the scaling with more caching distributions, placement policies, caching policy and so on.  Currently, this section only contains two figures which, in my opinion, do not present a sufficient comparison with other caching schemes and varying simulations setting. 

Author Response

See the pdf file.

Reviewer 2 Report

Overall, this is a very well written manuscript. I am not an expert in coded content delivery, but the proposed approach seems to be technically sound. The examples really help with improving the readability of the paper. The presentation is very nice, but I would only recommend adding a table (if space permits) for the notations used throughout the paper as a reference for the readers. 

However, the results section seems a bit thin, and it could easily be extended with additional simulation results. For example, it could be interesting to demonstrate how the tolerance of the traffic to delay impacts the performance and how much additional delay would be needed to aggregate the requests to improve the performance. This trade-off in performance vs additional delay could be interesting to see because it impacts the practicality of the approach. Also, the impact of heterogeneity in cache sizes and the run-times of the methods (in terms of computing the coloring and codes) could be interesting to present. 

Also, the experimental setting could be described slightly better. Details such as the simulator being used, the details of how the traffic is generated, etc. are missing.  

Minor thing:

- Spelling error in multiple places: factional -> fractional 

Author Response

See the PDF file. 
